# Associations of Diet Quality and Heavy Metals with Obesity in Adults: A Cross-Sectional Study from National Health and Nutrition Examination Survey (NHANES)

**DOI:** 10.3390/nu14194038

**Published:** 2022-09-28

**Authors:** Tiezheng Li, Luhua Yu, Zongming Yang, Peng Shen, Hongbo Lin, Liming Shui, Mengling Tang, Mingjuan Jin, Kun Chen, Jianbing Wang

**Affiliations:** 1Department of Public Health, National Clinical Research Center for Child Health, The Children’s Hospital, Zhejiang University School of Medicine, Hangzhou 310058, China; 2Department of Chronic Disease and Health Promotion, Yinzhou District Center for Disease Control and Prevention, Ningbo 315040, China; 3Department of Health Prevention, Yinzhou District Health Bureau of Ningbo, Ningbo 315040, China; 4Department of Public Health, Fourth Affiliated Hospital, Zhejiang University School of Medicine, Hangzhou 310058, China; 5Department of Public Health, Second Affiliated Hospital, Zhejiang University School of Medicine, Hangzhou 310058, China

**Keywords:** obesity, diet quality, heavy metal, cross-sectional study

## Abstract

A poor diet cannot fully explain the prevalence of obesity. Other environmental factors (e.g., heavy metals) have been reported to be associated with obesity. However, limited evidence is available for the combined effect of these factors on obesity. Therefore, we conducted a cross-sectional study and used the data from the National Health and Examination Survey (2007–2018) to explore the associations between diet quality and heavy metals and obesity. Diet quality was evaluated by the Healthy Eating Index-2015 (HEI-2015) score. Heavy metals included serum cadmium (Cd), lead (Pb), and mercury (Hg). We included 15,959 adults, with 5799 of obesity (body mass index ≥ 30 kg/m^2^). After adjustment for covariates, every interquartile range increase in HEI-2015 scores, Pb, Cd and Hg was associated with a 35% (odds ratios [OR] = 0.65, 95% confidence interval [CI]: 0.60, 0.70), 11% (OR = 0.89, 95% CI: 0.82, 0.98), 9% (OR = 0.91, 95% CI: 0.87, 0.96), 5% (OR = 0.85, 95% CI: 0.82, 0.89) reduction in risk of peripheral obesity, respectively. In addition, the association between the HEI-2015 scores and peripheral obesity was attenuated by higher levels of heavy metals (All *p* _interaction_ < 0.05). Results remained similar for abdominal obesity. Our study reveals the distinct effects of a high-quality diet and heavy metals on obesity prevalence, and the beneficial effect of a high-quality diet could be weakened by higher levels of heavy metals.

## 1. Introduction

Over the past five decades, the global prevalence of obesity has soared to a pandemic level [1,2], which has increased by 80% from 1980 to 2015 [3]. A study predicted that 57.8% of the world population would be overweight or obese by 2030 without effective intervention on current trends [3]. Many studies have identified obesity as a critical risk factor for all-cause mortality, metabolic diseases, cardiovascular diseases, musculoskeletal disease, Alzheimer’s disease, depression, and cancers in multiple organs (e.g., breast, ovarian, prostate, liver, kidney, and colon) [4,5]. Meanwhile, obesity is associated with unemployment, social disadvantages, and reduced socio-economic productivity [1]. 

High caloric diets and low physical levels are universally acknowledged as risk factors for obesity. Other factors might also exert effects on the risk of obesity [6]. Recent studies have suggested that environmental exposure, such as heavy metals [7], may play an important role in the onset of obesity. Some experimental evidence suggested that the hypothalamic dopaminergic system perturbation, endoplasmic reticulum stress, oxidative stress, impaired adipogenesis, and adipocytokines secretion might explain the associations between heavy metals and obesity [8,9,10,11]. However, the results from epidemiological studies are still controversial. A cross-sectional study conducted in the US suggested that barium (Ba) and thallium (Tl) were positively, while cadmium (Cd), cobalt (Co), cesium (Cs), and lead (Pb) were inversely associated with body mass index (BMI) and waist circumference (WC) [12]. However, the Korea National Health and Nutrition Survey reported that higher levels of blood Cd and mercury (Hg) were associated with a higher risk of obesity [13]. Moreover, a prospective cohort study observed no association of arsenic (As), Cd, Hg, Pb, or Co at baseline with BMI [14].

In the general non-smoking and non-occupationally exposed population, food is the most important source of heavy metals in the human body (e.g., Cd, Hg, and Pb) [15]. Metals are present in most foods at different concentrations as plant crops can absorb them from polluted soil or water and spread them through the food chain [16]. For example, fish, shellfish, and sea mammals are the main dietary sources of Hg, mainly methylated Hg (MeHg) [17]. Excessive heavy metals could accumulate in the body through food chains and disturb some essential nutrients’ normal absorption, distribution, or function [18].

Although numerous studies have investigated the association between diet and obesity, limited studies are available on the combined effect of heavy metals and diet on obesity. Herein, we aimed to explore the associations of heavy metals and diet quality with the risk of obesity and their potential interactive effects in this study.

## 2. Materials and Methods

### 2.1. Data Resource

NHANES is a nationally representative cross-sectional study, aiming to assess the health and nutritional information of adults and children in the United States (https://www.cdc.gov/nchs/nhanes/index.htm accessed on 20 January 2022). The NHANES used a stratified multistage probability cluster design and collected information on participants’ interviews (e.g., demographic, socioeconomic, dietary, and health-related questions) and laboratory tests (e.g., medical, dental, physiological measurements, and laboratory tests). The NHANES study protocol was reviewed and approved by the research ethics review board of the National Center for Health Statistics of the Centers for Disease Control and Prevention, and informed consent was obtained from all participants [19]. 

Considering the different measurement methods for some variables during the study period (e.g., diet, and physical activity), we used data from the six continuous NHANES cycles (i.e., 2007–2008, 2009–2010, 2011–2012, 2013–2014, 2015–2016, and 2017–2018). Among the 59,842 participants enrolled in NHANES 2007–2018, participants younger than 20 years (n = 25,072) and reported currently being pregnant (n = 372) were excluded. Moreover, we also excluded the participants without complete data on two 24-h dietary recalls (n = 1294), blood concentrations of heavy metals (n = 6109), BMI (n = 1829) and WC (n = 1599), and other covariates (n = 7608). Finally, a total of 15,959 individuals were included in the final analysis. Figure 1 shows the detailed flowchart for inclusion/exclusion criteria for study subjects.

### 2.2. Diet Quality Scores 

Dietary intake was collected from the NHANES using two 24-h dietary recalls. The first 24-h recall interview was conducted in person in the Mobile Examination Center (MEC) by trained interviewers, and the second interview was performed by telephone or mail three to ten days later. Diet was assessed with the average of two 24-h dietary recalls, and diet quality was measured using the Healthy Eating Index-2015 (HEI-2015). HEI-2015 is the latest version of HEI, which can measure diet quality independent of quantity, assess alignment with the U.S. Dietary Guidelines for Americans (DGA) and monitor changes in dietary patterns [20]. HEI-2015 comprises 13 components, including nine adequacy components (total vegetables, greens and beans, total fruits, whole fruits, whole grains, dairy, total protein foods, seafood and plant proteins, and fatty acids) and four moderation components (sodium, refined grains, saturated fats, and added sugars) (Appendix A). The HEI-2015 scores ranged from 0 to 100, with a higher score indicating a better quality of overall diet [20]. 

### 2.3. Heavy Metal Measurements

The methodological details of the laboratory analyses have been described on the NHANES website. Briefly, whole blood Pb, Cd, and Hg concentrations were measured by the Centers for Disease Control and Prevention’s National Center for Environmental Health (NCEH) [19]. Pb, Cd, and Hg levels were determined by inductively coupled plasma dynamic reaction cell-mass spectrometry (ICP-DRC-MS). Metal concentrations below the limit of detection (LOD) were imputed using the LOD divided by the square root of two. 

### 2.4. Obesity

Anthropometric parameters were measured by trained health technicians at the MEC following the standard protocols [19]. Standing height was measured to the nearest 0.1 cm, body weight was measured to the nearest 0.1 kg, and WC was measured to the nearest 0.1 cm. BMI was calculated as weight in kilograms divided by height in meters squared. Peripheral obesity was defined as BMI ≥ 30 kg/m^2^, and abdominal obesity was defined as WC ≥ 102 cm in males and ≥88 cm in females [21].

### 2.5. Covariates

Data on covariates included demographic characteristics, individual-level socioeconomic status (SES), lifestyle factors, and chronic health conditions. Demographic characteristics included age (continuous in years), sex (males and females), race/ethnicity (non-Hispanic white, non-Hispanic Black, Hispanic, and others), marital status (married/living with a partner, divorced/widowed/separated, and single/never married). Individual-level SES included education level and family income. Education level was categorized as less than high school, high school or General Educational Development (GED), and college or above. Family income was divided into three levels, including <1.30, 1.30 ≥ & < 3.49, and ≥3.50, by the ratio of family income to poverty (FPL) [22]. Lifestyle factors were obtained by questionnaire, including smoking (current, former, and never), alcohol consumption status (never, light, moderate, and heavy), and physical activity (insufficient activity and recommended activity). Current smokers were defined as individuals who smoked at least 100 cigarettes in their lifetime and currently smoking. Former smokers were defined as having more than 100 cigarettes in their lifetime but did not smoke at the time of the interview [23]. Alcohol intake was calculated based on self-reported drinking frequency (days per week, month, or year) and drinking quantity in the past year, and was categorized into four groups, including never (0 drink per week), light (<1 drink per week), moderate (1 ≥ & < 8 drinks per week), and heavy (≥8 drinks per week) [24]. According to World Health Organization recommendations on physical activity for health, physical activity was categorized into insufficient activity (<150 min of moderate-intensity activity each week, <75 min of vigorous-intensity activity, and less than an equivalent combination) and recommended activity (≥150 min of moderate-intensity activity each week, ≥75 min of vigorous-intensity activity, or greater than or equal to an equivalent combination) [25,26]. Chronic health conditions, including diabetes and cardiovascular disease (congestive heart failure, congestive heart failure, angina/angina pectoris, heart attack, or stroke), were measured in the “diabetes” and “Medical Conditions” questionnaires, respectively.

### 2.6. Statistical Analysis 

Baseline characteristics of study participants were presented as percentages for categorical variables and mean ± standard deviation (SD) for normally distributed variables or median (P_25_, P_75_) for non-normally distributed variables. The differences in characteristics between obesity and non-obesity were tested using Student’s *t*-test for normal data, Wilcoxon rank-sum test for non-normal data, and Chi-square test for categorical variables. All blood metal concentrations (Pb, Hg, and Cd) were log_10_-transformed due to right-skewed distribution and presented as geometric means (GMs) and geometric standard deviation (GSD). Coefficients of correlations between heavy metals (Pb, Hg, and Cd) and the components of the HEI-2015 scores were calculated and presented via a correlation-matrix heat map.

We used weighted logistic regression models to estimate odds ratios (ORs) and 95% confidence intervals (CIs) for the associations of HEI-2015 scores and blood heavy metals with the risk of obesity. Covariates were included in the following models: Model 1 adjusted for age, sex, ethnicity, education level, family income, and marriage; Model 2 further adjusted for smoking status, alcohol consumption, and physical activity; Model 3 further adjusted for prevalence of diabetes and cardiovascular disease. 

Stratified analyses on the associations between HEI-2015 scores and obesity were performed by age (20–29, 30–44, 45–64, and ≥65 years), sex (males and females), race/ethnicity (non-Hispanic white, non-Hispanic Black, Hispanic, and others), education level (less than high school, high school or GED, and college or above), marital status (married/living with partner, divorced/widowed/separated, and single/never married), family income (<1.30, 1.30–3.49 and ≥3.50), smoking status (current, former, and never), alcohol consumption status (never, light, moderate, and heavy), physical activity (insufficient activity and recommended activity), diabetes (yes and no), and cardiovascular disease (yes and no). For heavy metals, subgroup analyses were performed by age, sex, smoking status, and alcohol consumption status. Potential interactions were tested by including a multiplicative interaction term in the regression models. Moreover, interactive effects between heavy metals and HEI-2015 scores on the risk of obesity were explored by calculating the ORs for the associations between HEI-2015 scores and obesity in different quartiles of heavy metals.

Several sensitivity analyses were performed in our study to assess the robustness of the results. Firstly, participants with diabetes or cardiovascular disease were excluded since they might change their lifestyles and lose weight. Secondly, we repeated the analyses using data from the eight continuous NHANES cycles during 2003–2018.

All tests were two-sided, and *p* values less than 0.05 were considered statistically significant. All analyses accounted for the complex survey design and NHANES probabilistic sampling weights using R software (version 3.6.3) with the “survey” package.

## 3. Results

Overall, a total of 15,959 participants were included in the final analysis, and the mean age was 45.71 (±16.31) years, and 47.27% were females. About 36.3% of participants had peripheral obesity, and 53.4% had abdominal obesity. Mean concentrations of blood Cd, Pb, and Hg were 0.32 (±1.40), 1.06 (±1.26), and 0.92 (±1.64), respectively. The mean score of HEI-2015 was 53.97 (±13.62). In general, obese participants were more likely to be older, smokers, nondrinkers, divorced, Non-Hispanic Black, have lower education, lower family income, or lower physical activity as compared with normal-weight participants (Table 1).

As shown in Figure 2, a number of dietary components were associated with blood levels of heavy metals (Pb, Cd, and Hg). Especially, higher refined grain and sodium consumption was correlated with Cd (*p* < 0.001). Vegetables, refined grains, sodium, and saturated fats were correlated with significantly higher levels of blood Pb (*p* < 0.001). In addition, Hg was correlated with seafood and plant protein, added sugar, total fruits, and vegetable intake (*p* < 0.001). 

Table 2 shows the multivariate-adjusted ORs and 95% CIs for the associations of HEI-2015 scores and heavy metals with adiposity indicators. In the full adjusted model, a higher HEI-2015 score was associated with a lower risk of peripheral obesity (Quartile 4 vs. Quartile 1: OR = 0.47, 95% CI: 0.41, 0.54) and abdominal obesity (Quartile 4 vs. Quartile 1: OR = 0.51, 95% CI: 0.45, 0.57). Moreover, all of the three heavy metals were inversely associated with risk of peripheral obesity (Quartile 4 vs. Quartile 1: OR = 0.48, 95% CI: 0.40, 0.57 for Pb; OR = 0.47, 95% CI: 0.39, 0.59 for Cd; OR = 0.57, 95% CI: 0.49, 0.67 for Hg) and abdominal obesity (Quartile 4 vs. Quartile 1: OR = 0.55, 95% CI: 0.46, 0.65 for Pb; OR = 0.50, 95% CI: 0.42, 0.60 for Cd; OR = 0.56, 95% CI: 0.49, 0.65 for Hg).

In stratified analyses, the associations of HEI-2015 scores with peripheral and abdominal obesities did not vary across the subgroups by age, alcohol consumption status, physical activity, race/ethnicity, diabetes, and cardiovascular disease (All *p* _interaction_ > 0.05, Figure 3). However, stronger associations were observed among females, nonsmokers, and individuals with higher education, higher income, and married/living with partners (All *p* _interaction_ < 0.05, Figure 3). For heavy metals, stronger associations were observed among women and older participants (All *p*
_interaction_ < 0.05, Figure 4). Interaction analysis for HEI-2015 scores and heavy metals showed that higher concentrations of heavy metals could attenuate the beneficial effect of healthy dietary patterns (higher diet scores) on the risk of peripheral or abdominal obesity. However, the association between HEI-2015 in the highest quartile and obesity was not offset by heavy metals, even in their highest quartiles. (Figure 5).

Sensitivity analysis excluding subjects with diabetes or cardiovascular disease did not materially change our results (Appendix A). Moreover, sensitivity analysis using NHANES subsets from 2003 to 2018 obtained similar results for the associations of HEI-2015 scores and heavy metals with obesity (Appendix A). 

## 4. Discussions

Our study was based on a large population and showed that higher HEI-2015 scores were significantly associated with lower peripheral or central obesity prevalence. Stronger associations were observed among females, nonsmokers, and individuals with higher education, higher income, and married or living with a partner. Similar associations were observed for heavy metals (Pb, Cd, and Hg) and obesity. We also found interactive effects between heavy metals and diet scores on the risk of obesity.

Previous studies have established that one of the leading causes of obesity is positive energy balance, in which energy intake exceeds energy expenditure. In addition, dietary components play crucial roles in metabolism and energy balance [27]. For example, a low-carbohydrate, high-fat diet could reduce insulin secretion, increase fat mobilization from adipose tissue, and stimulate free fatty acids’ oxidation, thus increasing body fat loss and energy expenditure [28,29]. High-protein diet could maintain basal body fat, prevent fat-free mass loss, increase satiety, produce thermal effects, and lead to weight loss [30]. The HEI-2015 score was commonly used to measure dietary patterns in previous studies [20]. Higher HEI-2015 scores are similar to the “Mediterranean-like” dietary pattern characterized by higher consumption of fruits, vegetables, whole grains, white meats, fish, and monounsaturated fats, which could reduce the risk of oxidative stress and inflammation [31,32]. However, lower HEI-2015 scores are similar to the “calorie-dense” pattern characterized by high intakes of refined grains, starches, desserts, sweets, red meats, alcohol, and saturated fats, leading to higher calorie intakes [32]. Our results showed that higher HEI-2015 scores were significantly associated with a lower risk of obesity, consistent with previous studies. A previous study using NHANES III data found that the HEI-1995 score was associated with a lower risk for abdominal obesity in US adults [33]. Inverse associations of HEI-2005 with BMI and WC were observed in a longitudinal study based on the Multi-Ethnic Study of Atherosclerosis (MESA) [34]. However, a study in Brazil reported no relationship of HEI with BMI and WC [35]. A possible explanation of the discrepancies may be that HEI, designed for US populations, is a measure for assessing whether a food set aligns with the DGA [36] and may not be suitable for dietary assessment in other populations [35,37]. 

We also observed more substantial beneficial effects of diet quality for obesity among women, non-smokers, and participants who were married to/lived with a partner, had higher education and higher income, or had more physical activity. A Canadian Community Health Survey revealed a stronger association between diet quality and obesity in women [38]. A study on the health effects of overweight and obesity in 195 countries over 25 years suggested the prevalence of obesity was higher in women than in men at all sociodemographic levels [39]. Meanwhile, individuals’ positions in the social hierarchy (such as educational attainment, household income, and neighborhood deprivation) shape their access to health-promoting resources and their exposure and vulnerability to adverse environmental conditions [40,41]. Furthermore, a systematic review showed that in most cases, subjects who adhered to diet quality indices had favorable health behaviors associated with being older, married, higher education levels, and lower smoking [42].

In the present study, we found that heavy metals (Pb, Cd, and Hg) were inversely associated with risks of peripheral or abdominal obesity, and this finding was consistent with previous studies. For example, a cross-sectional study using NHANES data from 1999 to 2002 observed that Pb and Cd were inversely associated with BMI and WC [12]. Another study using the NHANES data during 2007–2010 reported that blood Hg was also inversely associated with BMI in adults [43]. However, the mechanism for the relationship between heavy metals and obesity remains unclear. Some animal studies indicated that Pb could reduce weight by perturbating the hypothalamic dopaminergic system [8]. As an important endocrine disruptor, Hg has been indicated to have a potential role in the pathogenesis of obesity [44]. For example, animal experiments showed that HgCl_2_ treatment significantly decreased serum leptin levels with the down-regulation of leptin mRNA expression in white adipose tissue and reduced adipocyte size [9]. Cd may affect body weight by increasing oxidative stress, affecting adipose tissue and glucose metabolism [45]. 

Our study also observed stronger associations between heavy metals and obesity among women and older participants. Women are more susceptible to heavy metals than men due to the different redox homeostasis processes, hormonal influences (sexual maturation and menopause), and immune responses between the sexes [46,47]. Moreover, behavioral factors, such as smoking and occupational exposure, might affect the different susceptibility between men and women. For example, smokers could be less sensitive to heavy metals contained in tobacco [48]. Women are more susceptible to the adverse effects of heavy metals than men due to the lower smoking rate in women. Compared with young people, the elderly may have higher concentrations of heavy metals, due to their accumulative effect on tissues and organs [49].

We found that heavy metals (Pb, Hg, and Cd) were associated with intakes of specific food groups. Our results were consistent with previous studies analyzing the elemental content of these foods and their contribution to human exposure. The FDA’s Total Diet Study, a comprehensive assessment of heavy metals in U.S. food sources, reported that Pb was primarily from cucumbers, sweet peaches, shrimp, cattle liver, and fruit cocktails; Hg was mainly from canned shrimp, tuna casserole, tuna, haddock, and fish sticks; and Cd comes mainly from beef liver, spinach, and iceberg lettuce [50]. A cross-sectional study suggested that fish and seafood were the sources of the largest quantity of dietary Hg exposure, and the source of non-seafood products (e.g., vegetables, and rice) could not be ignored [17]. Meanwhile, a study showed that cereals and breads, followed by leafy vegetables including potatoes, were the top food groups contributing to Americans’ dietary Cd intake, accounting for 66% of the total estimated dietary Cd [51]. Collectively, due to food being an important source of heavy metals, it is critical for us to reduce exposure by lowering the safety threshold in foods [52]. We also found interactive effects between heavy metals and diet on the risk of obesity. High concentrations of heavy metals could attenuate the beneficial effect of healthy dietary patterns on obesity. Previous studies reported that trace elements and vitamins rich in high-quality dietary patterns were positively associated with healthy body weight. For example, zinc (Zn) and related protein families are involved in adipocyte metabolism, which plays a crucial role in controlling energy balance [53]. Vitamin A and vitamin A-binding proteins play essential roles in the metabolic process, such as adipocyte differentiation, adipogenesis, and lipid metabolism. Dietary vitamin A supplementation administration has been recommended to prevent the development of obesity [54]. Our study also found inverse associations between HEI-2015 scores and the risk of obesity. However, we noticed that the beneficial effect of HEI-2015 scores could be attenuated in subjects with higher serum heavy metals. Possible explanations could be that high heavy metals in the body might interfere with the normal functional performance of nutrients. Previous studies have reported that a high Hg concentration could block Zn intake [55], and Cd might prevent vitamin A from releasing into the blood [56]. However, we also noticed that the offsetting effect of heavy metals was more significant when the diet had a lower score. In other words, people with a less healthy diet should pay more attention to the effect of heavy metals. Further studies are needed to explore the relationship between nutrients and heavy metals in the metabolic process. 

Obesity, as one of the most serious global health challenges, could impair health and quality of life and increase the burden on the healthcare system [57]. Therefore, urgent action is needed to prevent and control obesity. The results of our study could provide new evidence on obesity management strategies and policies to better control and prevent obesity. However, several limitations should be considered in our study. Firstly, the study design was cross-sectional, and the exposures and outcomes were surveyed simultaneously, it may be difficult to establish the temporal sequence between exposures and outcomes. Therefore, the results cannot conclude a cause–effect relationship, and reverse causality could not be ruled out. Secondly, the HEI-2015 score was calculated based on self-reported 24-h dietary recall data, and participants were subject to over- or under-reporting. Thirdly, although we controlled for several potential confounders in models, it should be noted that other unmeasured factors, such as workload, family history of diseases, and medication, could affect the results when extrapolating our results to other populations. Finally, we only considered the effect of single metals on obesity and ignored the actual scenario of exposure to multiple metals on obesity.

## 5. Conclusions

In summary, in this population-based cross-sectional study, higher diet quality was associated with lower abdominal and peripheral obesity risks. Stronger associations were observed among women, nonsmokers, or participants who were married, high-educated, or high-income earners. We also found that heavy metals (Cd, Pb, and Hg) were inversely associated with lower abdominal and peripheral obesity risks. Additionally, the beneficial effect of higher diet quality on the risk of obesity could be attenuated by higher levels of heavy metals (Pb, Hg, and Cd). Our findings highlighted that stakeholder, including environmental regulators, public health experts, legislators, hygiene managers, and hygiene supervisors for food, need to work together to enhance the quality of people’s diets, and take heavy metals into consideration when during dietary management to control body weight since heavy metals might counteract the beneficial effect of healthy dietary patterns on obesity.

## Figures and Tables

**Figure 1 nutrients-14-04038-f001:**
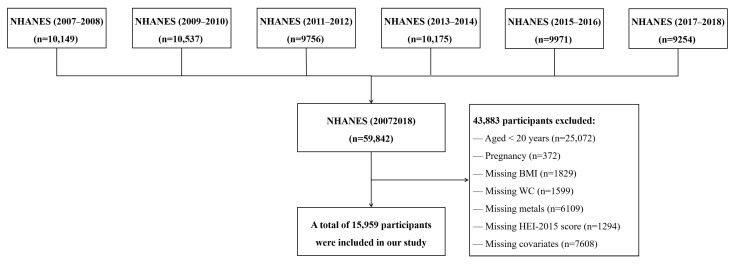
Flowchart of the study population.

**Figure 2 nutrients-14-04038-f002:**
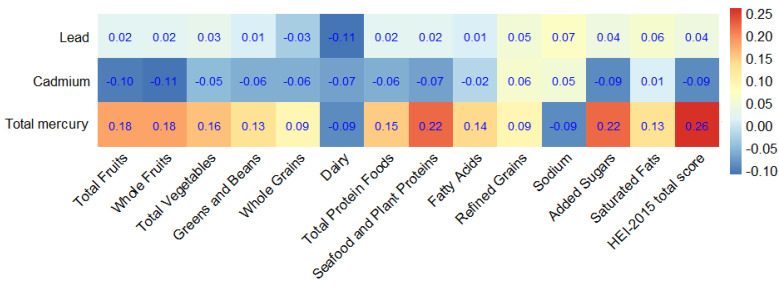
Spearman correlations between heavy metals and the components of the HEI-2015 total score.

**Figure 3 nutrients-14-04038-f003:**
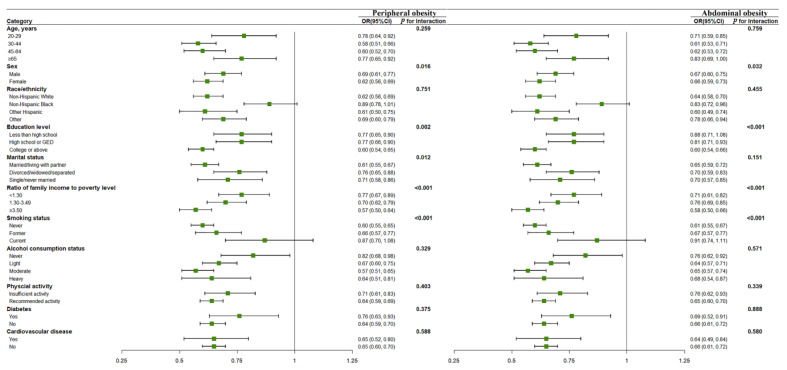
Subgroup analyses for the associations of HEI-2015 score (per interquartile range increment) with risk of obesity. Note: NHANES, National Health and Nutrition Examination Survey; BMI, body mass index; WC, waist circumference; HEI, The Healthy Eating Index. All results were adjusted for age, sex, race, education level, marital status, income, smoking, alcohol consumption, physical activity, diabetes, and cardiovascular disease, except for stratified variables separately.

**Figure 4 nutrients-14-04038-f004:**
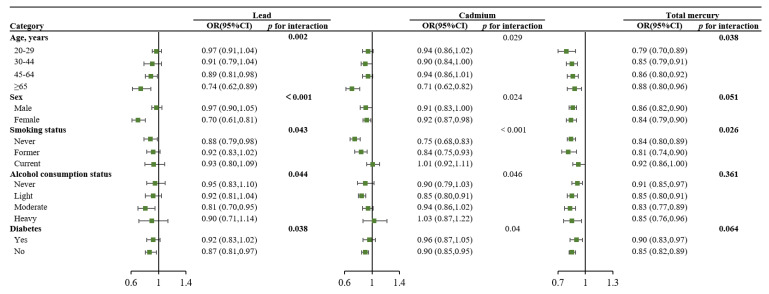
Subgroup analyses for the associations of heavy metals (per interquartile range increment) with risk of obesity. Upper figure: Subgroup analyses for heavy metals and risk of peripheral obesity. Lower figure: Subgroup analyses for heavy metals and risk of abdominal obesity. NHANES, National Health and Nutrition Examination Survey; BMI, body mass index; WC, waist circumference; HEI, The Healthy Eating Index. All results were adjusted for age, sex, race, education level, marital status, income, smoking, alcohol consumption, physical activity, diabetes, and cardiovascular disease, except for stratified variables separately.

**Figure 5 nutrients-14-04038-f005:**
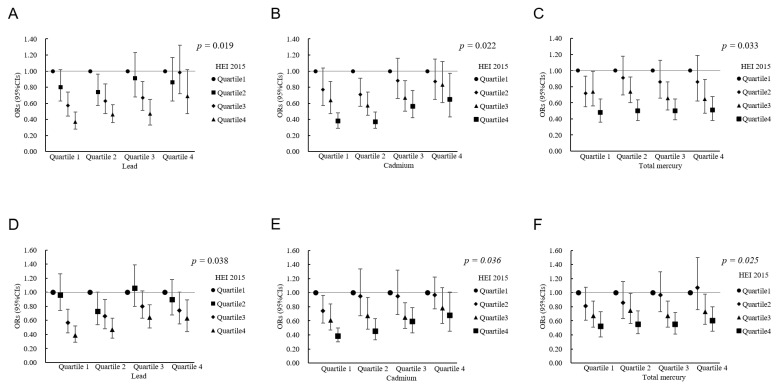
The association between HEI 2015 and obesity in different quartiles of heavy metals in blood. Note: (**A**–**C**) The association between HEI 2015 and peripheral obesity in different quartiles of lead, cadmium, and total mercury; Peripheral obesity was defined as BMI ≥ 30 kg/m^2^. (**D**–**F**) The association between HEI 2015 and abdominal obesity in different quartiles of lead, cadmium, and total mercury. Abdominal obesity was defined as a WC of ≥102 cm for males and ≥88 cm for females.

**Table 1 nutrients-14-04038-t001:** Demographic characteristics of the NHANES 2007–2018 participants (*n* = 15,959).

Characteristics	Total (*n* = 15,959)	No Obesity (*n* = 10,160)	Obesity (*n* = 5799)	*p* Value
Age (years), mean (SD)	45.71 (16.31)	45.27(16.74)	46.53 (15.46)	0.004
Sex, *n* (%)				0.509
Male	8415 (52.73)	5582 (54.94)	2833 (48.85)	
Female	7544 (47.27)	4578 (45.06)	2966 (51.15)	
Race/ethnicity, *n* (%)				<0.001
Non-Hispanic White	7240 (45.37)	4822 (47.46)	2418 (41.70)	
Non-Hispanic Black	3164 (19.83)	1657 (16.31)	1507 (25.99)	
Other Hispanic	1509 (9.46)	967 (9.52)	542 (9.35)	
Others	4046 (25.35)	2714 (26.71)	1332 (22.97)	
Education level, *n* (%)				<0.001
Less than high school	3231 (20.25)	2030 (19.98)	1201 (20.71)	
High school graduate or GED	3584 (22.46)	2151 (21.17)	1433 (24.71)	
College or above	9144 (57.30)	5979 (58.85)	3165 (54.58)	
Marital status, *n* (%)				<0.001
Married/living with partner	9648 (60.45)	6154 (60.57)	3494 (60.25)	
Divorced/widowed/separated	3129 (19.61)	1876 (18.46)	1253 (21.61)	
Single/never married	3182 (19.94)	2130 (20.96)	1052 (18.14)	
Ratio of family income to poverty level, *n* (%)				0.005
<1.30	4738 (29.69)	2957 (29.10)	1781 (30.71)	
1.30–3.49	5936 (37.20)	3670 (36.12)	2266 (39.08)	
≥3.50	5285 (33.12)	3533 (34.77)	1752 (30.21)	
Smoking status, *n* (%)				<0.001
Never	8844 (55.42)	5599 (55.11)	3245 (55.96)	
Former	3818 (23.92)	2286 (22.50)	1532 (26.42)	
Current	3297 (20.66)	2275 (22.39)	1022 (17.62)	
Alcohol consumption status, *n* (%)				<0.001
Never	2429 (15.22)	1422 (14.00)	1007 (17.37)	
Light	6566 (41.14)	3984 (39.21)	2582 (44.52)	
Moderate	5102 (31.97)	3465 (34.10)	1637 (28.23)	
Heavy	1862 (11.67)	1289 (12.69)	573 (9.88)	
Physical activity, *n* (%)				<0.001
Insufficient activity	2853 (17.88)	1698 (16.71)	1155 (19.92)	
Recommended activity	13,106 (82.12)	8462 (83.29)	4644 (80.08)	
Diabetes, *n* (%)				<0.001
Yes	2013 (12.61)	832 (8.19)	1181 (20.37)	
No	13,946 (87.39)	9328 (91.81)	4618 (79.63)	
Cardiovascular disease, *n* (%)				<0.001
Yes	1318 (8.26)	730 (7.19)	588 (10.14)	
No	14,641 (91.74)	9430 (92.81)	5211 (89.86)	
BMI (kg/m^2^), mean (SD)	28.63 (6.51)	24.88 (3.10)	35.60 (5.34)	<0.001
Waist circumference (cm), mean (SD)	98.21 (16.08)	89.64 (10.17)	114.13 (12.52)	<0.001
HEI-2015 total score, mean (SD)	53.97 (13.62)	55.17 (13.89)	51.74 (12.82)	<0.001
Cadmium (μg/L), GM (GSD)	0.32 (1.40)	0.33 (1.41)	0.30 (1.38)	<0.001 *
Lead (μg/dL), GM (GSD)	1.06 (1.26)	1.11 (1.25)	0.96 (1.26)	<0.001 *
Total mercury (μg/L), GM (GSD)	0.92 (1.64)	0.99 (1.67)	0.80 (1.52)	<0.001 *

Note: NHANES, National Health and Nutrition Examination Survey; BMI, body mass index; WC, waist circumference; HEI, The Healthy Eating Index. Data are presented as mean ± SD, Geometric mean (GM) ± geometric standard deviation (GSD), or n (%). The *t*-test and χ2 test were between the peripheral obesity and no obesity groups. * Wilcoxon rank-sum test was used for non-normal distribution data. Obesity was defined as BMI ≥ 30 kg/m^2^.

**Table 2 nutrients-14-04038-t002:** Association of HEI-2015 total scores and heavy metals with obesity in NHANES 2007–2018 (*n* = 15,959).

Exposure	Peripheral Obesity ^a^			Abdominal Obesity ^b^		
	Model 1 ^c^ OR (95% CI)	Model 2 ^d^ OR (95% CI)	Model 3 ^e^ OR (95% CI)	Model 1 ^c^ OR (95% CI)	Model 2 ^d^ OR (95% CI)	Model 3 ^e^ OR (95% CI)
HEI-2015 total score						
Quartile 1 ^f^	1.00 (Ref)	1.00 (Ref)	1.00 (Ref)	1.00 (Ref)	1.00 (Ref)	1.00 (Ref)
Quartile 2	0.84 (0.73, 0.95)	0.82 (0.71, 0.93)	0.81 (0.70, 0.93)	0.91 (0.78, 1.05)	0.89 (0.76, 1.03)	0.88 (0.75, 1.03)
Quartile 3	0.69 (0.60, 0.79)	0.67 (0.58, 0.77)	0.67 (0.58, 0.77)	0.67(0.59, 0.79)	0.66(0.57, 0.76)	0.66 (0.57, 0.77)
Quartile 4	0.49 (0.44, 0.56)	0.48 (0.42, 0.54)	0.47 (0.41, 0.54)	0.53 (0.46, 0.60)	0.51(0.45, 0.58)	0.51 (0.45, 0.57)
*P* for trend ^g^	<0.001	<0.001	<0.001	<0.001	<0.001	<0.001
Continuous (per IQR)	0.67 (0.62, 0.71)	0.65 (0.61, 0.70)	0.65 (0.60, 0.70)	0.67 (0.63, 0.73)	0.66 (0.62, 0.71)	0.66 (0.62, 0.71)
Pb						
Quartile 1 ^f^	1.00 (Ref)	1.00 (Ref)	1.00 (Ref)	1.00 (Ref)	1.00 (Ref)	1.00 (Ref)
Quartile 2	0.79 (0.68, 0.91)	0.81 (0.70, 0.93)	0.83 (0.72, 0.96)	0.83 (0.72, 0.95)	0.83 (0.72, 0.96)	0.85 (0.74, 0.98)
Quartile 3	0.54 (0.46, 0.62)	0.57 (0.49, 0.66)	0.62 (0.54, 0.72)	0.68 (0.59, 0.79)	0.69 (0.59, 0.81)	0.74 (0.64, 0.87)
Quartile 4	0.39 (0.33, 0.47)	0.42 (0.35, 0.50)	0.48 (0.40, 0.57)	0.49 (0.41, 0.57)	0.49 (0.41, 0.59)	0.55 (0.46, 0.65)
*P* for trend ^g^	0.001	0.004	0.013	<0.001	<0.001	<0.001
Continuous (per IQR)	0.83 (0.75, 0.93)	0.86 (0.78, 0.95)	0.89 (0.82, 0.98)	0.87 (0.82, 0.93)	0.88 (0.83, 0.94)	0.90 (0.86, 0.96)
Cd						
Quartile 1 ^f^	1.00 (Ref)	1.00 (Ref)	1.00 (Ref)	1.00 (Ref)	1.00 (Ref)	1.00 (Ref)
Quartile 2	0.84 (0.74, 0.96)	0.83 (0.73, 0.94)	0.85 (0.75, 0.97)	0.86 (0.76, 0.99)	0.83 (0.73, 0.95)	0.85 (0.74, 0.97)
Quartile 3	0.69 (0.60, 0.81)	0.66 (0.57, 0.78)	0.70 (0.60, 0.82)	0.75 (0.64, 0.87)	0.69 (0.59, 0.81)	0.72 (0.61, 0.84)
Quartile 4	0.51 (0.45, 0.59)	0.45 (0.38, 0.54)	0.47 (0.39, 0.57)	0.59 (0.52, 0.68)	0.48 (0.40, 0.57)	0.50 (0.42, 0.60)
*P* for trend ^g^	<0.001	<0.001	<0.001	<0.001	<0.001	<0.001
Continuous (per IQR)	0.90 (0.86, 0.94)	0.91 (0.87, 0.95)	0.91 (0.87, 0.96)	0.93 (0.89, 0.96)	0.91 (0.88, 0.95)	0.92 (0.88, 0.96)
Hg						
Quartile 1 ^f^	1.00 (Ref)	1.00 (Ref)	1.00 (Ref)	1.00 (Ref)	1.00 (Ref)	1.00 (Ref)
Quartile 2	0.92 (0.81, 1.04)	0.92 (0.81, 1.04)	0.93 (0.82, 1.06)	0.99 (0.85, 1.14)	0.99 (0.85, 1.14)	1.00 (0.86, 1.16)
Quartile 3	0.83 (0.72, 0.96)	0.85 (0.73, 0.98)	0.88 (0.75, 1.03)	0.79 (0.67, 0.93)	0.79 (0.67, 0.94)	0.81 (0.69, 0.96)
Quartile 4	0.53 (0.45, 0.62)	0.55 (0.47, 0.65)	0.57 (0.49, 0.67)	0.54 (0.46, 0.62)	0.55 (0.47, 0.63)	0.56 (0.49, 0.65)
*p* for trend ^g^	<0.001	<0.001	<0.001	<0.001	<0.001	<0.001
Continuous (per IQR)	0.84 (0.80, 0.87)	0.85 (0.81, 0.88)	0.85 (0.82, 0.89)	0.85 (0.82, 0.89)	0.86 (0.83, 0.89)	0.86 (0.83, 0.89)

Note: HEI, The Healthy Eating Index; CI, confidence interval; OR, odds ratio; IQR, interquartile range; Pb, Lead; Cd, Cadmium; Hg, Total mercury. ^a^ Peripheral obesity was defined as BMI ≥ 30 kg/m^2^. ^b^ Abdominal peripheral obesity was defined as a WC of ≥102 cm for males and ≥88 cm for females. ^c^ Adjusted for demographics characteristics, including sex, age, race, income, education, and marriage. ^d^ Adjusted for covariates in model 1+smoking, drinking status, and physical activity.^e^ Adjusted for covariates in model 2+diabetes and cardiovascular disease. ^f^ Least HEI-2015 total score quartile. ^g^
*P* values for trend were derived based on ordinal quartile values.

## Data Availability

Publicly available datasets were analyzed in this study. These data can be found here: [https://wwwn.cdc.gov/nchs/nhanes/ accessed on 20 January 2022].

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
