# Peer review of "Associations of Diet Quality and Heavy Metals with Obesity in Adults: A Cross-Sectional Study from National Health and Nutrition Examination Survey (NHANES)"

_nutrients, 2022, doi:10.3390/nu14194038_

Round 1
Reviewer 1 Report
This manuscript summarizes a study of the independent and joint associations of diet quality and heavy metals with obesity. They used data from the US National Health and Examination Survey (NHANES) from 2007-2018 to measure obesity according to both BMI and waist circumference, dietary quality from the Healthy Eating Index–2015 (HEI–2015) score, and levels of Pb, Hg, and Cd in whole blood. The risk of obesity was lower among those with higher diet quality and with higher blood concentrations of each heavy metals. The benefit of a better diet, however, was somewhat attenuated by higher levels of heavy metals.
The use of NHANES is a strength, and the sample size is large. For the most part, the methods are appropriate, and the manuscript is well written. My major concerns are listed below:
· A very surprising findings is that the heavy metals help protect against obesity. In fact, the individual ORs for each quartile of Hg, Cd, and Pb shown in Table 2 are similar to or in some cases stronger than those for HEI-2015. It appears, however, that the authors are minimizing this finding. For example, only a single paragraph addresses this finding in the Discussion, and it is completely omitted in both the Abstract and Conclusions. Much more attention needs to be devoted to this finding and to possible explanations. For example, were levels of the metals associated with the HEI-2015 and its components such as consumption of fish or fruits and vegetables?
· I believe that the authors overemphasize the interactions between the metals and the HEI-2015. The finding that the interactions were statistically significant reflects the large sample size. The magnitude of the attenuating effects, however, appear to be small. In fact, the ORs for quartile 4 of the HEI-2015 increase only slightly with increasing metal levels and remain statistically significant for even the highest quartile of each metal. This contradicts their conclusion “heavy metals might counteract the beneficial effect of healthy dietary patterns on obesity” (lines 337-338). The finding that heavy metals are protective and that even high levels only slightly attenuate the effects of a healthy diet also weakens the argument that the study has important implications for policy makers (lines 333-336).
· More information regarding why blood was used instead of urine for measuring exposure, especially for Hg and Cd, and its limitations. For example, blood Hg is largely organic methyl Hg, while urinary Hg measures inorganic forms. Similarly, blood Cd is more reflective of current exposure whereas urinary Cd measures cumulative exposure.
· The limitations of the study are only simplistically discussed (lines 320 – 327). For example, what are the problems with cross-sectional studies, and why can they not be used to evaluate causality? Has the validity of using self-reported dietary recall for the HEI-2015 score been evaluated? What unmeasured factors may have confounded the results? Were levels of the metals positively correlated? If yes, why were they only considered singly and not jointly?
Reviewer 2 Report
The research methodology is reasonable and the results are appropriately presented. However, there is a lack of discussion on the relationship between the Healthy Eating Index-2015 (HEI-2015) diet and heavy metals. Mercury and lead are ingested from fish, Lead is ingested from vegetables, and these food consist of HEI-2015score. Please add a discussion of these associations.
Author Response
Please check the revised manuscript using the “Track Changes” function.
Point 1: The research methodology is reasonable and the results are appropriately presented. However, there is a lack of discussion on the relationship between the Healthy Eating Index-2015 (HEI-2015) diet and heavy metals. Mercury and lead are ingested from fish, Lead is ingested from vegetables, and these foods consist of HEI-2015score. Please add a discussion of these associations.
Response 1: Thanks for the reviewer’s comment. We apologize for our unclear description for the association between heavy metals (Pb, Cd and Hg) and the risk of obesity in the Abstract (lines 31-32) and Conclusions (lines363-365). As suggested, we have now modified these sections in the current revised manuscript. Meanwhile, we calculated the Spearman correlations to explore the associations of heavy metals with intakes of specific food groups, described and discussed the association in the Results and Discussion (Figure 2, lines 200-205, and lines 310-323).